# Revealing the Sequence Characteristics and Molecular Mechanisms of ACE Inhibitory Peptides by Comprehensive Characterization of 160,000 Tetrapeptides

**DOI:** 10.3390/foods12081573

**Published:** 2023-04-07

**Authors:** Mingzhe Ma, Yinghui Feng, Yulu Miao, Qiang Shen, Shuting Tang, Juan Dong, John Z. H. Zhang, Lujia Zhang

**Affiliations:** 1Shanghai Engineering Research Center of Molecular Therapeutics & New Drug Development, School of Chemistry and Molecular Engineering, East China Normal University, Shanghai 200062, China; 52204300067@stu.ecnu.edu.cn (M.M.); yhfeng@chem.ecnu.edu.cn (Y.F.); ylm695@163.com (Y.M.); 13661657406@163.com (Q.S.); 2School of Food Science and Technology, Shihezi University, Shihezi 832000, China; 15729932216@163.com (S.T.); dongjuanvv@126.com (J.D.); 3NYU-ECNU Center for Computational Chemistry at NYU Shanghai, Shanghai 200062, China; 4Department of Chemistry, New York University, New York, NY 10003, USA

**Keywords:** angiotensin-converting enzyme (ACE), ACE inhibitory peptides, tetrapeptides, molecular docking, molecular dynamics simulations

## Abstract

Chronic diseases, such as hypertension, cause great harm to human health. Conventional drugs have promising therapeutic effects, but also cause significant side effects. Food-sourced angiotensin-converting enzyme (ACE) inhibitory peptides are an excellent therapeutic alternative to pharmaceuticals, as they have fewer side effects. However, there is no systematic and effective screening method for ACE inhibitory peptides, and the lack of understanding of the sequence characteristics and molecular mechanism of these inhibitory peptides poses a major obstacle to the development of ACE inhibitory peptides. Through systematically calculating the binding effects of 160,000 tetrapeptides with ACE by molecular docking, we found that peptides with Tyr, Phe, His, Arg, and especially Trp were the characteristic amino acids of ACE inhibitory peptides. The tetrapeptides of WWNW, WRQF, WFRV, YYWK, WWDW, and WWTY rank in the top 10 peptides exhibiting significantly high ACE inhibiting behaviors, with IC_50_ values between 19.98 ± 8.19 μM and 36.76 ± 1.32 μM. Salt bridges, π–π stacking, π–cations, and hydrogen bonds contributed to the high binding characteristics of the inhibitors and ACE. Introducing eight Trp into rabbit skeletal muscle protein (no Trp in wide sequence) endowed the protein with a more than 90% ACE inhibition rate, further suggesting that meat with a high content of Trp could have potential utility in hypertension regulation. This study provides a clear direction for the development and screening of ACE inhibitory peptides.

## 1. Introduction

Hypertension [1] is a chronic disease that is widely present in adults of middle to high age, and as the condition worsens, it can lead to further cardiovascular disease [2], kidney disease [3], stroke [4], etc. It is reported that more than 25% of the world’s population suffers from hypertension [5]. Blood pressure is regulated by a number of metabolic pathways, the most important of which are the renin–angiotensin system (RAS) [6] and the kallikrein–kinin system (KKS) [7]. Angiotensin-converting enzyme (ACE) [8] is one of the key regulators of these two systems, and is an important target for the study of antihypertensive drugs [9,10]. The main treatment for chronic conditions is medication, but unlike acute conditions, the side effects of powerful drugs may be more harmful than the chronic disease itself.

The antihypertensive drugs currently used in clinical practice have different degrees of side effects, such as captopril [11] and ramipril [12], which are ACE inhibiting drugs, causing irritating cough, angioedema, and hypotension while regulating blood pressure [13]. Therefore, the development of food-derived antihypertensive peptides with high safety, wide sources, and low cost has attracted much attention [14]. Peptides with ACE inhibiting properties have been reported in milk [15], soy [16], peas [17], fish [18], and reishi [19]. Among them, short-chain peptides [20] such as dipeptides, tripeptides, and tetrapeptides are not easily redigested by digestive enzymes, and can be absorbed into the bloodstream better and faster [21]. Therefore, short-chain peptides are the key targets for ACE inhibiting peptide research.

Pan et al. [22] used molecular docking to investigate the mechanism of ACE inhibition by Leu-Leu, a hydrolysate from whey protein. Sun et al. [23] used quantitative structure–activity relationship (QSAR) modeling to find that short-chain peptides with more hydrogen bonding acceptor groups at the C-terminus are prone to form hydrogen bonds with ACE, thereby stabilizing the complex structure. Yan et al. [24] used QSAR modeling to study the characteristics of ACE inhibitory peptides and found that for tripeptides, the N-terminal is a large group with a positively charged residue in the middle, and a hydrophobic residue at the C-terminal, which facilitates ACE inhibition, while zinc ions are important for the stability of the ACE complex. These studies show that it is feasible to apply molecular simulation techniques to the study of ACE inhibitory peptides, but the lack of systematic elaboration of the characteristics of ACE inhibitory peptides, and the explanation of the inhibition mechanisms, etc., make it difficult to provide research directions for the discovery of new inhibitory peptides.

By evaluating the inhibitory effect of 8000 tripeptides, our group, Chen et al. [25], showed that potential ACE inhibitory peptides prefer aromatic amino acids at both ends of the peptide, especially at the C-terminus, where Trp-containing tripeptides may have better ACE inhibitory effects. This result is in general agreement with the findings of Panyayai et al. [26] for tripeptides. However, the screening and studying of ACE inhibitory peptides with only tripeptides is inadequate, as 8000 represents only a very small proportion of these short-chain peptides, dipeptides, tripeptides, and tetrapeptides. Moreover, the relatively small sample size makes it difficult to detect systematic characteristic patterns, so it is necessary to study more short-chain peptides. No systematic research of the potential inhibitory effect of 160,000 tetrapeptides on ACE have been conducted.

In this study, the ACE inhibitory effects of 160,000 tetrapeptides were systematically evaluated by molecular docking for the first time, and Tyr, Phe, His, Arg, and especially Trp were found to be the characteristic amino acids of ACE inhibitory peptides by statistical analysis. Molecular dynamics simulations, binding free energy calculations, interaction relationship analysis, in vitro IC_50_ values determination, and validation of ACE inhibitory activity of the hydrolysed rabbit-derived protein indicated that peptides containing Trp, Tyr, Phe, His, Arg, and especially Trp at the C- and N-terminal ends, had high ACE inhibitory activity. Therefore, we have successfully established an efficient and accurate screening strategy for ACE inhibitory peptides. The sequence characteristics of the ACE inhibitory tetrapeptide were systematically characterized for the first time, and proved to be correct by calculations and experiments. This study provides a clear direction for the development and screening of ACE inhibitory peptides.

## 2. Materials and Methods

### 2.1. Structure of ACE and Tetrapeptide Complexes

First, the Protein Preparation Wizard module in Schrödinger [27,28] was used to process the structure of ACE (PDB ID: 1O8A), keeping the Zn (II) in the system and deleting the rest of the small molecules, and then the optimised structure was obtained through the OPLS_2005 force field. A Python script was used to generate a sequence of 160,000 tetrapeptides, a ChemScript script was called to convert the tetrapeptides into a 3D conformation, and then the Ligprep module in Schrödinger was used to optimise the structure in the OPLS3 force field to obtain a reasonable 3D conformation of the low-energy tetrapeptide. Finally, 160,000 tetrapeptides were individually docked to ACE by the Glide module in a semi-flexible docking strategy, with Zn (II) as the centre to generate the ACE–tetrapeptide complex conformation.

### 2.2. Tetrapeptide Sequence Characterization Analysis

Tetrapeptides were selected for association analysis by RapidMiner Studio version 7.4. In order to fit association rule studies, physical properties of the amino acids were assigned as a dataset. The data (CSV file format) were imported to workflow processing by using RapidMiner’s import operator. The main process in this workflow contains FP-growth and creates association rules, which algorithms run on Rapidminer. The prediction association rules were evaluated by using support and confidence as the main filtering parameters in the system. The strength of an association rule can be measured in terms of its support and confidence. Association rules are created by analysing data for frequent if/then patterns, and using the criteria support and confidence to identify the most important relationships. Support is an indication of how frequently the items appear in the database. Confidence indicates the number of times the if/then statements were found to be true. The formal definitions of these metrics are:$$\mathrm{Support},\;s(X\to Y)=\frac{Sup(XUY)}{number\;of\;transactions}$$
$$\mathrm{Confidence},c\;(X\to Y)=\frac{Sup(XUY)}{Sup(X)}$$

### 2.3. MD Simulation Process

The structure of the complex generated by Schrödinger docking was subjected to parallel MD simulations to investigate the binding mechanism of ACE and the tetrapeptide. The cationic dummy atom (CaDA) [29,30] was used to process the Zn (II) in the protein and generate the corresponding parameter files. The FF99SB force field was used to generate the parameter files for ACE, and the Leap in Amber18 module was used to fill in the missing atoms. The TIP3P water box model was used as the solvent environment, a buffer distance of 12 Å was set, and sodium ions were added to keep the system neutral. The steepest descent method and the conjugate gradient method were used to optimise the system for energy minimisation. Next, the system was ramped up to 300 K by maintaining a frequency of 1 ps rise of 1 K under a 10 kcal/(mol Å^2^) constraint. The SHAKE algorithm [31] was used to constrain the chemical bonding involving hydrogen atoms. Finally, 50 ns of MD simulations were performed at constant temperature and pressure without constraints. An average of 100 frames were extracted from the trajectories for enthalpy calculations, and an average of 10,000 frames for entropy calculations.

### 2.4. Calculation of Binding Free Energy

Calculation of the free energy of binding, consisting of enthalpy and entropy changes, was performed using the MM/GBSA and IE methods, respectively.

The MM/GBSA method, one of the most widely-used methods, can be expressed by the following equation:$$∆G_{bind}=∆G_{gas}+{∆G}_{solv}$$
where, ∆*G_gas_* and ∆*G_solv_* represent gas-phase energy and solvation free energy, respectively. Among them, ∆*G_gas_* consists of electrostatic interaction and van der Waals (vdW) interaction, as follows:$$∆G_{gas}=∆E_{ele}+∆E_{vdW}$$

∆*G_solv_* contains polar and nonpolar solvation free energy of two parts, as follows:$$∆G_{solv}=∆G_{gb}+∆G_{np}$$

The polar solvation free energy ∆*G_gb_* is calculated by the GB module, and the nonpolar solvation free energy ∆*G_np_* is calculated by:$$∆G_{np}=\gamma \times SASA+\beta$$

The *SASA* represents the solvent-accessible surface area, and is calculated by the MSMS program. The *γ* and *β* are set to 0.005 kcal·(mol∙Å^2^)^−1^ and 0.00 kcal·mol^−1^, respectively.

The IE formula is derived from the free energy of binding in the gas phase.
(1)$${∆G}_{gas}=-kTln\frac{\int {dq}_{w}dq_{p}dq_{l}e^{-\beta \left(E^{p}+E^{l}+E_{int}^{pl}+E^{w}+E_{int}^{pw}+E_{int}^{lw}\right)}}{\int {dq}_{w}dq_{p}dq_{l}e^{-\beta \left(E^{p}+E^{l}+E^{w}+E_{int}^{pw}+E_{int}^{lw}\right)}}\;=-kTln\left[\frac{1}{\langle e^{\beta E_{int}^{pl}}\rangle }\right]\;\;\;\;\;\;\;\;\;\;=-kTln\left[\frac{1}{\langle e^{\beta E_{int}^{pl}}\rangle }\right]\;\;\;\;\;\;\;\;\;\;\;\;\;\;=kTln\left[e^{\beta \langle E_{int}^{pl}\rangle }\langle e^{\beta \left(E_{int}^{pl}-\langle E_{int}^{pl}\rangle \right)}\rangle \right]\;\;\;\;\;\;\;\;\;\;=\langle E_{int}^{pl}\rangle +kTln\langle e^{\beta ∆E_{int}^{pl}}\rangle \;\;\;\;\;\;\;\;\;\;\;\;\;\;\;\begin{array}{c} =\langle E_{int}^{pl}\rangle -T∆S \end{array}\;\;\;\;\;\;\;\;\;\;\;\;\;$$
(2)$$-T∆S=kTln\langle e^{\beta ∆E_{int}^{pl}}\rangle$$

The contribution of electrostatic forces to the free energy of polar solvation is calculated using the GB model, where igb = 2.

### 2.5. Alanine Scan

All the amino acids on ACE within 5 Å of the tetrapeptide were mutated to alanine. ${∆G}_{bind}^{x}$ and ${∆G}_{bind}^{a}$ in Equation (3) represent the binding free energy of a residue before and after mutation to alanine, respectively. ${∆G}_{bind}$ in Equation (4) represents the total binding free energy, which is the sum of the energy contributions of the individual residues. AS is often used in conjunction with MM/GBSA and IE to calculate the binding free energy of receptors and ligands, and to analyse hot spot residues, referred to in this paper as the AS-GBIE method.
(3)$${∆∆G}_{bind}^{x\to a}={∆G}_{bind}^{x}-{∆G}_{bind}^{a}$$
(4)$${∆G}_{bind}=\sum_{x}{∆∆G}_{bind}^{x\to a}$$

### 2.6. Peptide Synthesis

Potential ACE inhibitory peptides, selected based on the docking results, were solid-phase synthesized from Bankpeptide Biological Technology Co., Ltd. The purity (>95%) of these peptides was verified by RP-HPLC on an Inertsil ODS-SP column (4.6 mm × 250 mm, 5 μm, Shimadzu Co. Ltd., Shanghai, China). Sequences were verified by mass spectrometry (Micromass ZQ, Waters, Shanghai, China).

### 2.7. Assay of ACE Inhibitory Activity

IC_50_ refers to the semi-inhibitory concentration; here, the principle of in vitro IC_50_ determination of the potential ACE inhibitory peptide is that ACE can hydrolyse HHL into marena and dipeptide. HHL and tetrapeptide are competitive, and when ACE binds to the tetrapeptide, it will reduce the production of the product marena; the IC_50_ activity of the tetrapeptide is when it is reduced by half of the original concentration. The water bath temperature was controlled at 37 °C, and the initial mass concentration of each tetrapeptide was 2 mg/L. The tetrapeptide was diluted with borate buffer solution into 7 concentration gradients, and coincubated with HHL buffer for 5 min. Then, the ACE solution was added to initiate the reaction, and the reaction was terminated by the addition of HCl after 30 min of system reaction. Afterwards, the reaction was analysed on a C18 column, with HPLC detection at 228 nm and an injection volume of 10 μL. The mobile phases were water and acetonitrile at 75% and 25%, respectively, with a flow rate of 1 mL/min, and isocratic elution for 7 min (acetonitrile and water containing 0.1% trifluoroacetic acid). ACE inhibition was calculated as ACE inhibition = ((b − c) − (a − c))/(b − c) × 100% (a: peak area of hippuric acid in the experimental group; b: peak area of hippuric acid in the blank group; c: peak area of hippuric acid in the control group), and the inhibition rate was converted into IC_50_ values by the software graphpad prism 7.0.

### 2.8. Vectors, Strains, Reagents, and Chemicals

The pET-22b (−) plasmid (Miaoling plasmid sharing platform, China) was used for subcloning *E. coli* DH5α and *E. coli* BL21(DE3) cells (Jierui Biotechnology Co., Ltd., Shanghai, China), which were used as the host for cloning, and for the expression of chymotrypsin and the mutants with higher Trp content. Primers were ordered by Personalbio Co., Ltd. (Shanghai, China). The plasmid extraction kits, Gel/PCR Purification Kit and Seamless Cloning Kit, were provided by Solarbio Science and Technology Co., Ltd. (Beijing, China). Substrate peptides were synthesized by ZhuanPeptide Inc. (Hangzhou, China). All other commercial chemicals were provided by the China National Pharmaceutical Group Corporation.

### 2.9. Heterologous Expression and Purification

The rabbit-derived skeletal myosin light chain C and its mutants with higher Trp content were expressed by *E. coli* BL21(DE3). The steps are as follows: (1) Single colonies of good growth on the transformed plates are picked and inoculated into 5 mL LB medium (containing 100 μg/mL ampicillin), and incubated for 12 h at 37 °C, 200 rpm. (2) Inoculate the culture in (1) with 1% of the inoculum into 50 mL LB medium (containing 100 μg/mL ampicillin), and incubate for 8 h at 37 °C, 220 rpm, in a baffled conical flask. (3) Inoculate the culture in (2) with 1% of the inoculum into 500 mL LB medium (containing 100 μg/mL ampicillin), and incubate in a baffled conical flask at 37 °C, 220 rpm, until OD600 = 0.8–1. (4) Add induction agent IPTG to (3) for a final concentration of 0.1 mM, induce at 20 °C, 200 rpm, for 12 h. (5) Take (4) and centrifuge at 8000× *g* for 5 min to collect the bacterium, and discard the supernatant. (6) Resuspend the bacterium in 0.9% saline, centrifuge again, discard the supernatant, repeat twice, resuspend with 50 mL PBS buffer (pH 7.4), and sonicate for 30 min (5 s sonication, 5 s interval). Centrifuge at 18,000× *g* and 4 °C for 10 min, and filter the supernatant by 0.22 μm filter membrane; the filtrate is used for subsequent purification. (7) Flush 10 column volumes with ultrapure water, flush 10 column volumes with equilibrium solution, top sample the crushed bacterial solution at a flow rate of 1 mL/min, and collect the perforated solution. (8) Rinse the nickel column sequentially with 25 mM imidazole buffer, 150 mM imidazole buffer, and 300 mM imidazole buffer, collect the eluate, and desalinate and concentrate the purified protein and store at −80 °C.

### 2.10. Determination of Inhibition of ACE Activity by Chymotrypsin Hydrolyse Rabbit Skeletal Muscle Actomyosin and Tryptophan Mutants

The purified rabbit skeletal muscle actomyosin and tryptophan mutants were dissolved in 0.1 M PBS buffer (pH 7.5). Proteins, at 5 mg/mL, with 25 μg/mL chymotrypsin at 37 °C for 2 h, were centrifuged at 20,000× *g* for 20 min, and filtered through a 0.22 μm filter. The optimum temperature for chymotrypsin was 30 °C. After hydrolysis for 2 h, the enzyme was inactivated by increasing the temperature to 95 °C, while the hydrolysis product remained stable at this temperature. The substrate HHL was broken down into hippuric acid (Hip) and the dipeptide His-Leu in the presence of ACE. The product hippuric acid showed maximum absorption at 228 nm, and was detected by HPLC on a C18 column. Incubation for 30 min at 37 °C was followed by the addition of 50 µL HCl to terminate the reaction. The hydrolysis products were first subjected to reductive alkylation, after which the treated samples were analysed by liquid–mass spectrometry (LC–MS/MS) to obtain the raw mass spectrometric results, which were analysed by the software Byonic to match the data and obtain the identification results.

## 3. Results and Discussion

### 3.1. Sequence Characterization of 160,000 Tetrapeptides Inhibition of ACE Activity

In this study, 160,000 rational conformations of tetrapeptides were created for the first time by Python and ChemScript scripts. Using Zn (II) in the ACE structure (PDB ID: 1o8a) as the docking centre, 160,000 complex structures of tetrapeptides with ACE were obtained, with scoring functions ranging from −12.125 to −1.536 (Appendix A). A lower score meant that the tetrapeptide binds more strongly to ACE, and might have a stronger inhibitory effect. There were two inflection points in the first 5000 and the last 5000, respectively, while the middle part showed a smoothing trend. This suggested that the first 5000 tetrapeptides may have a better inhibitory effect than the latter 5000 tetrapeptides. The residues composition at each site of the first 5000 and last 5000 tetrapeptides was performed (Figure 1). For the first 5000 peptides, the Trp appeared most frequently at the four sites, with percentages of 27.32%, 20.54%, 15.86%, and 20.86%, respectively (Figure 1A–D). The high composition of Trp at the N/C-terminal of the tetrapeptides was consistent with the results of Chen et al. and Panyayai et al. in the analysis of 8000 tripeptides. In addition to the high percentage of Trp, the residues of Tyr, Phe, Arg, and His were also highly present in the top 5000 peptides. This suggested that aromatic residues with high stocking characteristics, and the positive charged residues with potential for salt bridge or H bond forming, were significantly preferred for the inhibitors to better occupy the active pocket of ACE. For the last 5000 peptides, the amino acids at the four sites did not exhibit the same apparent residue preferences as the top 5000, but showed great nonpreference for the residues of Trp, Phe, Tyr, Arg, and His, with percentages less than 1%. The residues with short side-chains, such as Ile, Val, Ala, Thr, Cys, Leu, and Pro, were widely present in these last 5000 peptides, with percentage of 6.26–12.86% (Figure 1E–H). These statistics suggested that residues with short side-chains could not better recognize and anchor in the ACE pockets. The residue analysis of the ranked top 5000 and last 5000 residues suggested that the ACE may prefer peptides with big side-chains, such as aromatic or positively charged residues.

### 3.2. ACE Inhibitory Peptide Activity Determination

In vitro IC_50_ values (Table 1) showed that the top 10 tetrapeptides had IC_50_ values ranging from 19.98 ± 8.19 μM to 147.83 ± 14.27 μM, indicating that they all inhibited ACE to different degrees, with the best inhibitory effect being YYWK. The last two ranked tetrapeptides had IC_50_ values greater than 5000 μM in this assay, and had no significant inhibitory effect on ACE. Although the calculated results did not agree with those of the in vitro IC_50_ assay, they showed an overall consistent trend. WWNW, WRQF, WFRV, YYWK, WWDW, and WWTY inhibited ACE better than the other tetrapeptides, with IC_50_ values ranging from 19.98 ± 8.19 μM to 36.76 ± 1.32 μM, showing a stronger inhibitory effect on ACE. WFSW, WRFF, RWWD, and RRYQ had IC_50_ values ranging from 114.7 ± 6.15 μM to 147.83 ± 14.27 μM. The calculated results showed that the tetrapeptides with stronger binding capacities were WWNW, WRQF, WFRV, WWDW, WWTY, and WFSW, with binding free energy ranging from −27.35 kcal/mol to −20.31 kcal/mol. The results showed consistency between the calculation and the experiment. In addition, the second lowest binding free energy was calculated for YYWK, which was the best inhibited tetrapeptide in the experimental results, with a binding free energy of −18.27 kcal/mol. Although the calculations did not show an exact correlation with the experiments, the trends were generally consistent. This suggested that the virtual screening method and calculation strategy were valid, and tetrapeptides containing Trp, Tyr, Phe, His, or Arg might be potential ACE inhibitory peptides. In addition, the combining of docking score and binding free energy dual screening criteria might be more beneficial for the screening of dominant oligopeptides.

### 3.3. Mechanistic Analysis of ACE Inhibition of Tetrapeptides

To further investigate the mechanism of ACE inhibition by tetrapeptides, 2D interaction analysis was performed on the structures of the top 10 and the last 2 complexes of the docking score (Figure 2 and Appendix A). Most of the top-ranked tetrapeptides had more hydrogen bonds with Q281, H353, A356, and K511, and more hydrophobic interactions with Q281, H353, S355, H383, H387, H410, and H513. Some tetrapeptides (e.g., WWNW, WFRV) formed hydrogen bonds with some residues in the binding pockets of Appendix A (Ala354, Glu384, and Tyr523) and Appendix A (Gln281, His353, Lys511, His513, and Tyr520). A few tetrapeptides (e.g., WRQF) formed hydrogen bonds only with residues in Appendix A, while RWWD formed hydrogen bonds only with residues in Appendix A, but they formed more hydrogen bonds with residues outside the binding pocket, contributing to structural stability. The distance between Zn (II) (radius: 1.25 Å) and the oxygen (0.74 Å) or nitrogen (0.73 Å) atoms containing lone pairs of electrons in the adjacent tetrapeptide was close to the sum of their covalent radii, assuming a possible coordination interaction between Zn (II) and oxygen or nitrogen atoms. Zn (II) interacted with H383 (NE2), H387 (NE2), E411 (OE1), and tetrapeptides to form a distorted tetrahedral coordination that stabilized the complex conformation. In addition, most of the top 10 tetrapeptides also had π–π stacking effects with ACE, as well as forming hydrogen bonds with some residues outside the active site of ACE. These forces, together with hydrophobic interactions, contributed to the stability of the complex conformation, and enhanced the activity of the inhibitory peptide. Compared with the top-ranked peptides, the two last ranked tetrapeptides had fewer hydrogen bonds with ACE, and almost no hydrophobic interaction, which might account for the poor inhibition. Moreover, these tetrapeptides rarely contained aromatic amino acids, and had difficulty forming π–π stacking, which is not conducive to the stability of the complexes.

### 3.4. Analysis of the Binding Free Energy of Tetrapeptides Bound to ACE

In order to investigate the dynamic mechanism of ACE interaction with the tetrapeptides, and to analyse the possible reasons for the strength of the potential inhibitory peptide from an energetic point of view, molecular dynamics simulations were performed for 50 ns on the top 10 and last 2 structures of the ACE–tetrapeptide complexes. The RMSD values of the molecular dynamic simulations for the 12 complexes are shown in Appendix A. After 50 ns of MD simulations, all 12 complex systems reached stability, with RMSD values less than 2 Å. The binding free energy of the ACE–tetrapeptide complexes was calculated using the AS-GBIE method, and hot spot residues were disassembled (Appendix A). The binding free energies of the first ten and the last two ACE–tetrapeptide complexes are shown in Table 2. The contributions of electrostatic interactions and polar solvation free energy to the tetrapeptide binding of ACE were almost canceled out. The difference in −TΔS between the highest and lowest scoring tetrapeptides was small, but the difference in ΔH was large, with the largest contribution to ΔH being van der Waals interactions, resulting in high values of the total binding free energy for the lowest scoring tetrapeptides. The calculation of binding free energy clearly distinguished the highest and lowest scoring tetrapeptides, which was consistent with the docking results, indicating that the highest scoring tetrapeptides might be the ones with better ACE inhibition.

In order to investigate the hot spot residues in the binding of ACE to tetrapeptides, and to analyse the mechanism, an alanine scan was introduced to disentangle the ACE residues associated with binding. The amino acids contributing more than 1 kcal/mol to the overall binding free energy were defined as hot spot residues in this study, and the hot spot residues that appeared in the average of three parallel simulations of the top 10 scored ACE–tetrapeptide complexes are listed in Appendix A. A total of 17 hot spot residues contributing to the binding of the ten complexes were found, eight of which occurred with a probability greater than or equal to 50%, including Q281, H353, W357, E411, H513, Y520, R522, and Y523. This suggested that not only were the binding pocket and residues coordinated with Zn (II) essential for the binding of the tetrapeptide to ACE, but also some residues outside the binding pocket played a key role in the stability of the complex. All eight residues contributed less than −2 kcal/mol to one or more ACE–tetrapeptide complexes, especially Y523, which contributed −4.47 ± 0.30 kcal/mol to ACE–WWTY binding. To analyse the possible important role of these eight hot spot residues in the binding of the tetrapeptide by ACE, the framework conformation with the lowest potential energy from the MD trajectory was selected for interaction analysis using ACE–WWNW as an example (Appendix A). There were more hydrogen bonding interactions between these residues and WWNW, π–π stacking between W357 and WWNW, and π–cationic interactions between R522 and WWNW; these interactions might play an important role in stabilizing the conformation of ACE and tetrapeptide complexes. Half of these eight hot spot residues were charged amino acids, three of which were basic, and one was acidic. During the dynamic process of tetrapeptide binding to ACE, these charged residues might form salt bridges, π–π stacking, π–cations, and hydrogen bonds with the tetrapeptide, which facilitated the stabilization of the tetrapeptide within the binding pocket, and the inhibition of ACE. In contrast, tetrapeptides (such as GGSG and SGGG, which were smaller in size) might detach from the binding cavity during ACE binding, and have difficulty stabilizing inside. These tetrapeptides might be less prone to salt bridges, π–π stacking, etc., with ACE, which all have a stabilizing effect on receptor and ligand binding. To some extent, this qualitatively explained the phenomenon that the lower scoring tetrapeptides might have no inhibitory effect.

### 3.5. Experimental Validation of Rabbit-Derived Skeletal Muscle Globulin Hydrolysis

By sequence characterization of the ACE inhibitory activity of 160,000 tetrapeptides, the peptides with Trp at the C-terminus may have a better ACE inhibitory effect. Thus, this paper suggested that proteins containing a higher percentage of Trp may have a potential antihypertensive function. Among human gastrointestinal digestive enzymes, chymotrypsin tends to hydrolyse aromatic amino acids, and the hydrolysis of peptides by chymotrypsin could mimic the digestive environment of the human gastrointestinal tract for protein foods. In addition, the chymotrypsin P1 site is 93% selective for Trp, thus maximising the degradation of small peptides with terminal Trp in food-derived proteins. To demonstrate that Trp-rich food-derived proteins have better antihypertensive effects, the Trp-free rabbit-derived skeletal muscle myosin light chain C was studied in this paper, by introducing Trp sites on its α-helice and flexible loops exposed to solution, respectively. Mutant 1 (Mut 1) contained six Trp, and mutant 2 (Mut 2) contained eight Trp, and their sequences were compared in Appendix A. Chymotrypsin hydrolysed the rabbit-derived protein and its mutants, to achieve maximum hydrolysis and enrichment of small Trp-containing peptides. The potential antihypertensive function of the Trp-containing proteins was explored by comparing the inhibition rate of ACE activity by hydrolysis products (Figure 3A). The wild-type (WT) and mutants were expressed in *Escherichia coli,* and purified to obtain high purity proteins (Figure 3B and Appendix A). WT, Mut 1, and Mut 2 inhibited ACE activity by chymotrypsin hydrolysis, with inhibition rates of 27.07%, 36.56%, and 46.48%, respectively (Figure 3C). Considering that Mut 2 had only two more Trp than Mut 1, the 1.3% increase in Trp content was associated with the 9.92% increase in inhibition. Although it was not subjected to standard digestive tract simulations, this suggested a surprising result that the slight increase in the Trp content of the ingested protein was digested to produce a significant increase in the antihypertensive effect. The data from mass spectra (Appendix A) further demonstrated that a significant amount of small peptides with C-terminal Trp were present in the hydrolysis products. The higher inhibition rate of mutants with higher Trp content, after chymotrypsin hydrolysis, further demonstrated that small peptides with Trp at the C-terminus had significant ACE inhibitory activity. Finally, this experiment also demonstrated the feasibility of obtaining efficient ACE inhibitory peptides by protease-specific hydrolysis of food-derived proteins.

## 4. Conclusions

Food-derived ACE inhibitory peptides are excellent substitutes for highly effective drugs with many side effects. Efficient and accurate screening strategies, and knowledge of the sequence characteristics of inhibitory peptides and related mechanisms, are essential for the screening and development of ACE inhibitory peptides. In this paper, the low energy complex conformations of ACE with 160,000 tetrapeptides were systematically generated by molecular docking, and then, the ACE inhibitory effect of 160,000 tetrapeptides were preliminarily evaluated by a docking scoring function. The possible characteristics of the ACE inhibiting peptides were statistically analysed, and the presence of Trp, Tyr, Phe, His, or Arg residues at each site of the tetrapeptide had a strong ACE inhibitory effect, especially when Trp appeared at the C- and N-terminal. The top 10 and the last 2 ACE–tetrapeptide complexes were further analysed by molecular dynamics simulations, as well as MM/GBSA and IE methods. By calculating the binding free energy and analysing the possible reasons for the inhibitory effect of the top-ranked tetrapeptides containing these features, Van der Waals forces were found to play a major role in scoring the top-ranked complexes. The hot spot residue decomposition by the AS-GBIE method revealed that residues, including some binding pockets and some outside the binding pockets, contributed mainly to the binding of the tetrapeptides, including Q281, H353, W357, E411, H513, Y520, R522, and Y523. π–π stacking and π–cations also contributed to the stabilization of the structure. However, the tetrapeptides with a low docking score could not form most of the mentioned interactions that stabilize the complex conformation, and thus could not easily bind to ACE to form a stable complex for ACE inhibition.

The inhibition effects of the top 10 tetrapeptides and the last 2 tetrapeptides in the docking score were verified by in vitro IC_50_ assay. It was found that the top 10 tetrapeptides all had a significant inhibition effect on ACE, with different degrees of action. This result was also consistent with the calculations, suggesting that the results of the 160,000 tetrapeptide systematic docking analysis were valid. Potential ACE tetrapeptide inhibitory peptides may prefer amino acids Trp, Tyr, Phe, His, or Arg at each site. In vitro hydrolysis experiments of rabbit-derived skeletal myosin light chain C demonstrated that the antihypertensive capacity of the protein was activated with increasing Trp content in the sequence. Thus, Trp is extremely important in the sequence profile of ACE inhibitory peptides. Enzymatic hydrolysis of Trp-rich food-derived proteins is an important tool for developing ACE inhibitory peptides.

In summary, this article constructed a library of binding characteristics of all 160,000 tetrapeptides to ACE for the first time. Based on energy analysis, the sequence features of ACE inhibitory peptides were comprehensively and systematically elucidated. The molecular mechanism of ACE inhibitory peptide was clearly explained by calculation and experiment. This paper showed the direction for the screening and development of ACE inhibitory peptides.

## Figures and Tables

**Figure 1 foods-12-01573-f001:**
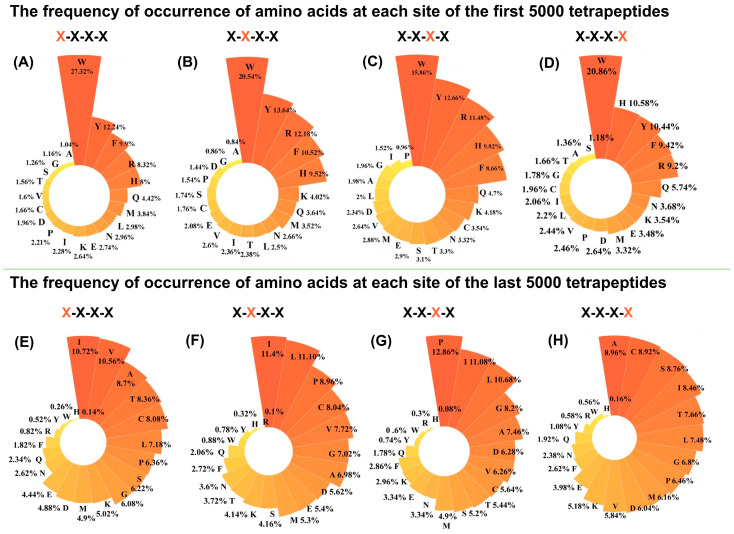
The frequency of occurrence of amino acids at each site of the first 5000 and the last 5000 tetrapeptides. The first 5000 tetrapeptides (**A**) N-terminal; (**B**) N-terminal second position; (**C**) N-terminal third position; (**D**) C-terminal. The last 5000 tetrapeptides (**E**) N-terminal; (**F**) N-terminal second position; (**G**) N-terminal third position; (**H**) C-terminal.

**Figure 2 foods-12-01573-f002:**
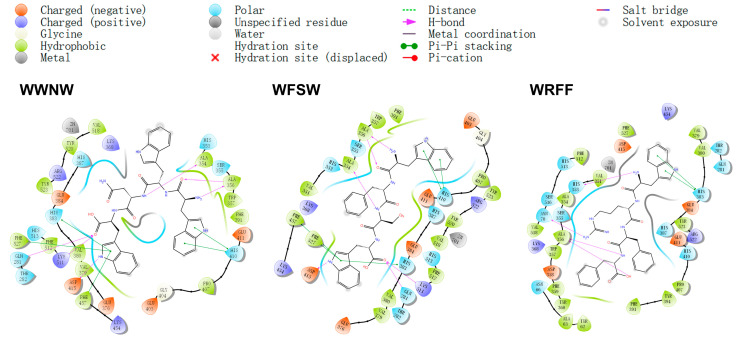
The 2D interaction map of ACE with the top three tetrapeptides.

**Figure 3 foods-12-01573-f003:**
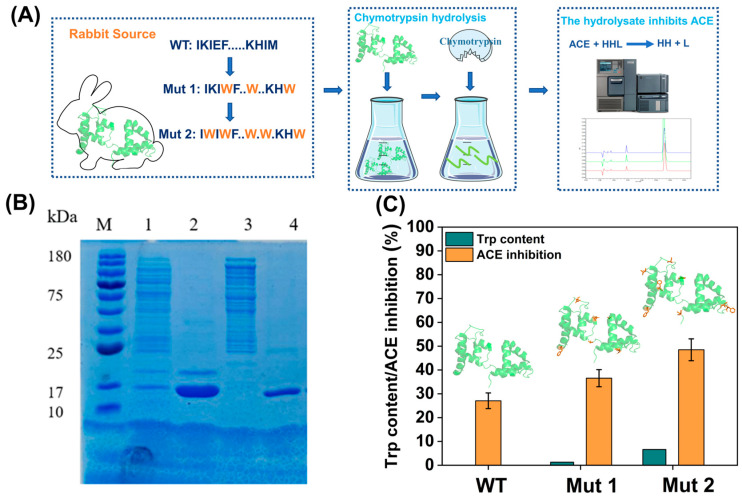
(**A**) Scheme of hydrolysis products inhibiting ACE activity. (**B**) Electrophoresis of purified proteins of mutants; 1–2: mut 1; 3–4: mut 2. (**C**) Effect of substrate Trp content on ACE inhibition rate.

**Table 1 foods-12-01573-t001:** Potential ACE inhibitory peptides IC_50_ values.

Docking Scoring Ranking	Tetrapeptide	IC_50_ (μM)
1	WWNW	26.44 ± 3.81
2	WFSW	147.83 ± 14.27
3	WRFF	140.93 ± 3.90
4	WRQF	29.58 ± 6.75
5	WFRV	31.75 ± 0.83
6	RWWD	128.90 ± 8.32
7	YYWK	19.98 ± 8.19
8	WWDW	36.76 ± 1.32
9	WWTY	28.50 ± 0.99
10	RRYQ	114.7 ± 6.15
15,999	GGSG	>5000
160,000	SGGG	>5000

**Table 2 foods-12-01573-t002:** Binding free energies of the top 10 and the last 2 ACE–tetrapeptide complexes calculated by the AS-GBIE method.

Docking Scoring Ranking	Complexes	$∆E_{vdW}$	$∆E_{ele}$	$∆G_{GB}$	$∆G_{np}$	∆H	−T∆S	$∆G_{bind}$
1	ACE-WWNW	−37.16	−32.50	33.11	−3.00	−39.54	12.19	−27.35
2	ACE-WFSW	−36.32	−43.52	41.12	−2.84	−41.56	19.86	−21.70
3	ACE-WRFF	−23.35	−38.23	41.48	−2.39	−22.50	13.12	−9.37
4	ACE-WRQF	−26.49	−58.36	55.04	−2.85	−32.66	12.36	−20.31
5	ACE-WRFV	−32.31	−52.88	49.07	−2.67	−38.78	13.68	−25.11
6	ACE-RWWD	−24.55	−50.01	50.40	−2.35	−26.60	20.79	−5.71
7	ACE-YYWK	−27.34	−49.40	49.47	−2.45	−29.72	11.45	−18.27
8	ACE-WWDW	−42.35	−7.87	14.14	−2.82	−38.90	17.98	−20.92
9	ACE-WWTY	−37.50	−42.12	39.43	−2.80	−42.98	16.53	−26.46
10	ACE-RRTQ	−28.42	−65.03	70.99	−2.45	−24.91	11.20	−13.71
159,999	ACE-GGSG	−12.60	−19.67	16.57	−1.39	−17.08	13.79	−3.29
160,000	ACE-SGGG	−14.51	16.93	−4.65	−1.26	−3.49	12.05	8.55

## Data Availability

Data is contained within the article or Appendix A.

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
