# Peer review of "Revealing the Sequence Characteristics and Molecular Mechanisms of ACE Inhibitory Peptides by Comprehensive Characterization of 160,000 Tetrapeptides"

_foods, 2023, doi:10.3390/foods12081573_

Round 1

Reviewer 1 Report

A very interesting and important work, presenting the results of research that can be a starting point for the search for ACE inhibitors, not only among new drugs but also food ingredients.

Hypertension is a chronic disease that is widely present in adults and it can lead to further cardiovascular disease, kidney disease, stroke, etc. More than 25% of the world's population suffers from hypertension. Angiotensin-converting enzyme (ACE) is one of the blood pressure-regulating proteins and an important target for the study of anti-hypertensive drugs. Captopril] and ramipril are ACE-inhibiting drugs. However, they are not without side effects, they include causing irritating cough, angioedema, and hypotension during regulating blood pressure

In a work entitled “Revealing the sequence characteristics and molecular mechanisms of ACE inhibitory peptides by comprehensive characterization of 160,000 tetrapeptides”, the authors, through systematically calculating the binding effects of 160,000 tetrapeptides with ACE by using molecular docking, were able to find that the peptide with Tyr, Phe, His, Arg and especially Trp are components of peptides that have the ability to inhibit ACE.

The most effective inhibitors are tetrapeptides of WWNW, WRQF, WFRV, YYWK, WWDW, and WWTY, with IC50 values between 19.98±8.19 26 μM and 36.76±1.32 μM. Additionally, it was shown that ionic interactions (salt bridges,) π-π stacking, π-cations, and hydrogen bonds are crucial to obtain an inhibitory effect. The authors showed that the following amino acid residues Q281, H353, W357, E411, H513, Y520, R522 and Y523 of ACE are crucial for interactions with inhibitors.

The results of the computational studies were also supported by data from in vitro studies, hydrolysis experiments of rabbit-derived skeletal myosin light chain C demonstrated that the anti-hypertensive capacity of the protein was activated with increasing Trp content in the sequence. The obtained results indicate that Trp is extremely important in the sequence profile of ACE inhibitory peptides. Enzymatic hydrolysis of Trp-rich food-derived proteins is an important tool for developing ACE inhibitory peptides.

However, the manuscript needs minor corrections, the most important include.

1) Figure 1 must be corrected, it is not legible in the presented form,

2) Information about the data presented in supporting materials must be completed (page 13)

3) There is no reference to Table S1 and Table S2 in the text of the manuscript.

Reviewer 2 Report

The authors report the identification of ACE inhibitor peptides (anti-hypertensive effects) using a combination of in silico and in vitro techniques. It is an article of interest for a general, and specialized audience.  amendments are required.

Recommendations:

It is recommendable to make complete and self-explanatory "figure legends" for clarity of the reader's comprehension

 Introduction:

Define LL; QSAR and all abbreviations that appear for the first time in the text.

Item 2.7. "Vector, Strains, reagents and chemicals" is not clear:

- ...cloning and expression for what (ACE or peptides?); substrate peptide?

 Item 2.8. Is myosin an enzyme? Or was the overexpression regarding ACE?

Define "PEP sample."

Results:

Is this section "Results and Discussion," or a deep discussion is lacking?

 define WT and other abbreviations in the text;

in item "3.2. ACE inhibitory peptide activity determination"…

Is the micromolar range of effect for an ACE inhibitor peptide acceptable compared with the peptides in the literature?

Discussion:

The authors mentioned that tryptophan is critical for the inhibitory effect of digested proteins that generate inhibitory ACE peptides.

Thinking about foods as the source of anti-hypertensive peptides, what kind of foods can naturally (without genetic modification) be sources of W-rich peptides with ACE inhibitory effects? Could you discuss more examples based on the present report? Could only meat provide such peptides? Or vegetal proteins also could?

Reviewer 3 Report

In this study, the authors reported the discovery of anti-ACE tetrapeptides by screening 160,000 tetrapeptides via molecular docking. Tyr, Phe, His, Arg and Trp were found to be the characteristic residues of anti-ACE peptides. Intermolecular interactions contributing to the binding of the tetrapeptides to ACE were reported. Ten top-ranked peptides were validated using in vitro anti-ACE assays. Trp-rich food proteins were suggested as desirable for the development of anti-ACE peptides. Overall, it is an interesting study and shortlisted in silico derived peptides were validated in wet lab assays.  I only have a few minor comments for the authors to consider during the revision process.

1.     The authors should explain why they chose the testicular ACE crystal structure (1O8A) rather than the somatic ACE crystal structure. Testicular ACE is confined to the testes, whereas somatic ACE can be found in the blood. The somatic ACE is therefore more relevant to the regulation of hypertension, which is the focus of this study.

2.     Did the authors perform a redocking step to validate their docking protocol?

3.  Reliable MD simulation results are essential to evaluate the dynamic interactions between ACE and the shortlisted peptides. Longer simulation times (at least 100 ns) and independent repeated simulations are required to draw meaningful conclusions.

4.     “E. coli” should be introduced in full the first time it is mentioned in the text. And because it is a scientific name, it should be in italics.

5.     The M&M section should include a section on statistical analysis, software/tools used, methods of analysis, number of replicates, etc.

6.     Heading in line 217 should be “3. Results and Discussion”.

7.     Please check again if the word "expelling" in line 237 is a typo? It seems out of place in the statement.

8.     Lines 238-244 - Is this statement consistent with previous observations in other studies?

9.     Lines 243-244 “the residues with less short chains could not exist in ACE for longer time” – this is confusing. Please rephrase.

10.  The results in Figure 1 are interesting. Have the authors tried to analyse a subset of 5000 tetrapeptides from the middle in Figure S1? Would it show a transition from W dominance (first 5000 peptides) to partial W dominance (middle 5000) and finally to lack of W dominance (last 5000 peptides)?

11.  Table 1 - Did the authors perform a proper correlation analysis to determine the relationship between the docking scores and the IC50 of the peptides? Or did the authors just do a visual check? Also, could the authors further explain why there was a lack of correction between the two parameters?

12.  There are online tools/servers for the prediction of anti-ACE peptides. Did the authors use these tools to support/complement their screening based on molecular docking?

13.  The authors only performed hydrolysis with chymotrypsin. My concern is that if a more complete GI digestion simulation were performed, would the same set of hydrolysis products (peptide fragments) be produced? In other words, how representative is the use of chymotrypsin alone?

14.  Another issue to consider: are the shortlisted tetrapeptides absorbable by the GI tract? Have the authors tried to predict this property using online/in silico tools?
